# Adherence to contemporary antiretroviral treatment regimens and impact on immunological and virologic outcomes in a US healthcare system

**Christophe T. Tchakoute**[1], **Soo-Yon Rhee**[2], **C. Bradley Hare**[3], **Robert W. Shafer**[2]*, **Kristin Sainani**[1]*

**1** Division of Epidemiology and Population Health, Department of Medicine, Stanford University, Stanford, CA, United States of America, **2** Division of Infectious Diseases, Department of Medicine, Stanford University, Stanford, CA, United States of America, **3** Department of Infectious Diseases, Kaiser Permanente Northern California, San Francisco, CA, United States of America

\* rwshafer@stanford.edu (RWS); kcobb@stanford.edu (KS)

**Data Availability Statement:** The data underlying this study are available at https://figshare.com/s/80775a12c838f6750541.

## Abstract

### Background

Only a few recent reports have examined longitudinal adherence patterns in US clinics and its impact on immunological and virological outcomes among large cohorts initiating contemporary antiretroviral therapy (ART) in US clinics.

### Methods

We followed all persons with HIV (PLWH) in a California clinic population initiating ART between 2010 and 2017. We estimated longitudinal adherence for each PLWH by calculating the medication possession ratio within multiple 6-month intervals using pharmacy refill records.

### Results

During the study, 2315 PWLH were followed for a median time of 210.8 weeks and only 179 (7.7%) were lost-to-follow-up. The mean adherence was 84.9%. Age (Hazard Ratio (HR): (95% confidence interval): 1.25 (1.20–1.31) per 10-year increase) and Black race (HR: 0.62 (0.53–0.73) vs. White) were associated with adherence in the cohort. A 10% percent increase in adherence increased the odds of being virally suppressed by 37% (OR and 95% CI: 1.37 [1.33–1.41]) and was associated with an increase in mean CD4 count by 8.54 cells/ul in the next 6-month interval (p-value <0.0001).

### Conclusions

Our study shows that despite large improvements in retention in care, demographic disparities in adherence to ART persist. Adherence was lower among younger patients and black patients. Our study confirmed the strong association between adherence to ART and viral

**Funding:** This study received support in part from the National Institute of Allergy and Infectious Diseases (NIAID) of the National Institute of Health (NIH) (award number AI136618, awarded to SYR and RWS). No additional external funding was received for this study.

**Competing interests:** The authors have read the journal's policy and have the following competing interests to declare: RWS previously received a research grant from Janssen Scientific Affairs (https://www.janssen.com/). This does not alter our adherence to PLOS ONE policies on sharing data and materials. There are no patents, products in development or marketed products associated with this research to declare.

suppression but could only establish a weak association between adherence and CD4 count. These findings reaffirm the importance of adherence and retention in care and further highlight the need for tailored patient-centered HIV Care Models as a strategy to improve PLWH's outcomes.

## Introduction

According to recent CDC estimates, about 1.2 million adults are currently living with HIV in the United States, with a large burden among men who have sex with men (MSM) [1]. The HIV care continuum has improved, and the availability of antiretroviral therapy (ART) has led to a decline in HIV-related mortality in the US [1, 2]. Adherence is now the main barrier to successful therapy [3–7] and is also a risk factor for the emergence of resistance [8–11].

However, while most studies on adherence have underscored the relationship between adherence and virologic failure, the impact of adherence on immunological recovery has not been well studied. In addition, there are only a few recent reports on longitudinal changes in CD4 count over an extended period among ART-naïve persons living with HIV (PLWH) in US clinic settings, and most studies assessing the impact of adherence on viral load had either relatively short follow-up durations or small sample sizes [8, 10, 12, 13].

ART adherence is challenging to evaluate in routine clinical settings. In such settings, medication possession ratio (MPR), calculated using pharmacy refill data, is the best proxy for adherence. This approach has been previously validated and used in many HIV cohort studies conducted in many settings [14–17]. In the current study, using pharmacy refill records, we first aimed to describe adherence levels in a dynamic population of ART-naïve PLWH initiating ART at an integrated US health care network between 2010 and 2017. We then identified baseline predictors that were associated with adherence as well as with loss to follow-up in the cohort. Finally, we investigated the association of adherence on future CD4 counts, as well as virus levels.

## Methods

### Study cohort

The study cohort comprised all PLWH in Kaiser Permanente Northern California (KPNC) undergoing first-line ART between January 2010 and December 2017 [18]. Demographic data, HIV-1 acquisition risk factors, virus load (VL), and CD4 counts were obtained from an electronic KPNC research database. Genotypic resistance testing was performed at the Stanford University Healthcare Clinical Virology Laboratory. ART history data were obtained from an electronic record of pharmacy pickups that recorded the date a prescription was filled and the duration of the prescription. All ART-naïve PLWH underwent HIV-1 reverse transcriptase (RT) and protease genotypic resistance testing. With few exceptions, ART was begun after the results of genotypic resistance testing were available. The Stanford University and KPNC institutional review boards approved this study. The KPNC research database were deidentified before use.

Retention in our cohort was high, with just 7.7% lost to follow-up (having no record of ART pickup or CD4 or VL testing for at least one year).

### Measurement of longitudinal adherence

Adherence was a repeated continuous outcome determined from the medication possession ratio (MPR). The MPR was defined as the percentage of time a PLWH possessed medication in each 6-month interval [14]. Each PLWH was assigned a start date based on the date they

filled their first ART prescription. Six-month intervals were then established beginning with this start date. We calculated the number of days that a PLWH was missing treatment in each 6-month interval as follows: (1) We calculated the number of days between pharmacy pickups; (2) We determined the number of daily doses of medication that a PLWH had available at the time of each pharmacy pick-up as the sum of the number of daily doses picked up plus any excess daily doses leftover from a previous pharmacy pick up; (3) If the daily doses available were less than the number of days between pharmacy pickups, we determined the specific dates on which a PLWH was missing treatment; (4) For each 6-month interval, we summed up the total number of missing treatment days that fell in the interval. Adherence for each 6-month interval was defined as: 100 * (number of days within an interval—number of missing treatment days) / number of days within the interval.

In addition to treating adherence as a continuous variable, adherence was also dichotomized into a binary repeated measures variable with two levels: optimal adherence defined as ≥95% adherence and suboptimal adherence defined as adherence <95%. The 95% threshold was chosen because in one of the early adherence studies, no deaths or opportunistic infections occurred among PLWH with an adherence level ≥95% on a protease inhibitor-containing regimen [19]. Hence the 95% adherence cut-off has been used since as a proxy for optimal adherence [20, 21].

## Viral load and CD4 counts

Between 2010 and 2014, VLs were measured using the VERSANT assay (Siemans Molecular Diagnostics), which has a range of 75–500,000 copies/ml. Between 2014 and 2018, VL was measured using the Ampliprep/Cobas Taqman assay 91 (Roche Laboratories), which has a range between 48–10 million copies/ml. In order to account for this change, we used 75 copies/ml and 500,000 copies/ml as the VL lower and upper limits, respectively. Both VL and CD4 counts were treated as continuous repeated measures variables.

## Clinical and demographic variables

Clinical and demographic variables collected at the time of ART initiation were: age, gender, ethnicity, HIV acquisition risk factors, baseline viral load, year of treatment initiation, transmitted drug resistance (TDR) (defined as the presence of one or more surveillance drug-resistance mutations prior to starting ART), and first-line treatment regimens.

## Statistical analyses

Descriptive statistics were used to summarize baseline demographic characteristics, HIV acquisition risk factors, baseline log10 viral load, CD4 counts, TDR prevalence, initial ART regimens, and adherence levels.

**a) GEE model to evaluate associations between baseline predictors and optimal adherence.** We first assessed the association between baseline characteristics and longitudinal adherence. Optimal adherence was defined as >95% adherence in each six month interval (as described above) and was treated as a binary repeated measure variable. To account for the correlated nature of the observations, we used univariable and multivariable Generalized Estimating Equations (GEE) with robust standard errors and a binomial distribution and logit link (as in logistic regression). A predictor was kept in the final model if it was significantly associated with longitudinal adherence or if it changed the odds ratio estimate of another variable that was significantly associated with the outcome by 10% or more. We included year of treatment initiation in all models as a potential confounder.

**b) GEE model to evaluate associations between longitudinal adherence and longitudinal CD4 counts/viral load.** We then examined the longitudinal association between adherence

and CD4 counts. We introduced a 6-month lag, whereby adherence in the previous 6-month interval was used to predict CD4 count in the next 6-month interval because CD4 responses upon treatment initiation have been shown to be delayed [22–24]. Adherence was modelled as a continuous repeated measures variable (percent adherent within a given 6-month interval). We used a GEE model with robust standard errors with a Gaussian distribution and identity link function. Graphical examinations of the data revealed that the relationship between CD4 counts and time was non-linear. To account for the non-linearity of time, a restricted cubic spline term with different knots was included for time. A similar approach was used to model the impact of adherence on log10 viral load (a continuous repeated measures variable).

**c) GEE models to evaluate temporal trends in CD4 counts and viral load by adherence level.** We also examined whether the relationship between time (predictor) and CD4 counts (repeated measures continuous variable) varied by adherence level. We fit a GEE model with CD4 count as the outcome and time as the predictor, as described in (b) above, but stratified by cumulative adherence groups rather than including adherence as a predictor in the model. The three adherence groups were cumulative adherence <70%, cumulative adherence between 70 and 95%, and cumulative adherence > = 95%. These percentages were chosen because based on the overall distribution of cumulative adherence in the population, similar proportions of patients in the data fell into each of these 3 categories. A similar approach was used for viral load treated as continuous repeated measures variable.

**d) GEE model to evaluate associations between longitudinal adherence and viral suppression.** We also examined the association between longitudinal adherence and viral suppression (a binary variable). Here, viral suppression—a binary repeated measures variable—was the outcome adherence (modelled as a continuous, repeated-measures variable) was the predictor. We used a GEE model with robust standard errors and a binomial distribution and logit link. As before, we used a six-month lag: Adherence within a given 6-month interval was used to predict viral suppression (undetectable viral load) in the next 6-month interval.

## Results

### Characteristics of the cohort

A total of 2315 PLWH initiated ART between January 2010 and December 2017. The median age at treatment initiation was 39 years. 91% of participants were men and 320 (13.8%) patients had TDR (Table 1). The median CD4 count and median VL at baseline were 373 cells/mm3 (IQR: 201 cells/mm3-537 cells/mm3) and 4.5 log10 copies/ml (IQR:4.0 log10 copies/ml -5.1 log10 copies/ml), respectively. 41.5% of PLWH were White, non-Hispanic. MSM comprised 60.4% of the cohort (Table 1).

### Adherence among PLWH at KPNC

The 2315 PLWH contributed to 98534 pharmacy refill records. The median number of 6-month intervals per person was 9 (IQR:5–14). The mean level of adherence per person was 84.9%. 57.6% of PLWH had an average adherence level ≥ 95% over the entire follow-up duration (Fig 1).

Table 2 shows the baseline predictors that were associated with optimal adherence, defined as at least 95% adherence in a given 6-month interval. After controlling for year of treatment initiation, four baseline variables remained significantly associated with optimal adherence: age, ethnicity, transmission risk factor, and first-line regimen. A ten-year increase in age was associated with a 25% increase in odds of optimal adherence (Adjusted odds ratio and 95% CI per ten-year increase: 1.25 [1.22–1.31]). Among ethnicities, Blacks had significantly lower odds of optimal adherence when compared with Whites (Adjusted OR and 95% CI: 0.62 [0.53–0.73]). In addition, injecting drug users were also less likely to achieve optimal adherence when compared with the

**Table 1. Baseline characteristics of the cohort at start of antiretroviral therapy, median (IQR) or N (%).**

| Characteristics | N = 2315 |
|---|---|
| Age (Median & SD) | 39 (12.7) |
| O percentile | 17 |
| 25th percentile | 29 |
| 50th percentile | 39 |
| 75th percentile | 49 |
| 100th percentile | 79 |
| Birth Sex: Male (%) | 90.8 |
| Ethnicity: | 960(41, 5%) |
| White, Not Hispanic (n, %) | |
| Hispanic(n, %) | 531 (22.9%) |
| African American (n, %) | 497 (21.5%) |
| Asian/ Pacific Islanders (n,%) | 234 (10,1%) |
| American Indians & Others (n, %) | 93 (4%) |
| Transmitted drug resistance (%) | 13.8% |
| CD4 count (cells/mm$^3$) (Median &IQR) | 373 (201–537) |
| <100 (n,%) | 330 (14.3) |
| 100–200 (n,%) | 232(10) |
| 200–350 (n,%) | 468(20.2) |
| 350–500 (n,%) | 563(24.3) |
| ≥ 500 (n,%) | 668 (28.9) |
| Log10 Viral Load (Median & IQR) | 4.5 (4.0–5.1) |
| Absolute Viral Load (copies/ml) (n, %) | 112(4.8) |
| < 1000 | |
| 1000–10 000 | 389 (16.8) |
| 10 000–100 000 | 1086(46.9) |
| 100 000–500 000 | 485(21) |
| > = 500 000 | 197(8.5) |
| Unknown (%) | 46(1.9) |
| HIV transmission Risk Factor (n,%) | 1398 (60.4) |
| MSM | |
| Male Bisexual | 275 (11.9) |
| Male Heterosexual | 242 (10.5) |
| Female Heterosexual | 175 (7.6) |
| Injection drug users | 128 (5.5) |
| Other/Unknown | 97 (4.2) |

MSM reference group (Adjusted OR and 95% CI: 0.77 [0.59–0.99]) (Table 2). Finally, among first-line regimen anchor drugs, only dolutegravir and raltegravir-based regimens differed significantly from the reference–efavirenz-based regimens (ORs [95%CI]: 1.35 [1.05–1.74] & 1.50 [1.20–1.87], respectively). We also looked at adherence rates according to regimen classes and observed that PLWH on integrase strand transfer inhibitors (INSTIs) were more likely to achieve optimal adherence when compared with PLWH on Non-nucleoside reverse transcriptase inhibitors (NNRTIs) (Adjusted OR and 95% CI: 1.21 (1.05–1.40).

## Effect of adherence on immunological outcomes

A total of 23470 CD4 counts were observed in the 2315 PLWH included in the final analysis. Over the entire follow-up duration, each patient had a median of 9 CD4 count measurements

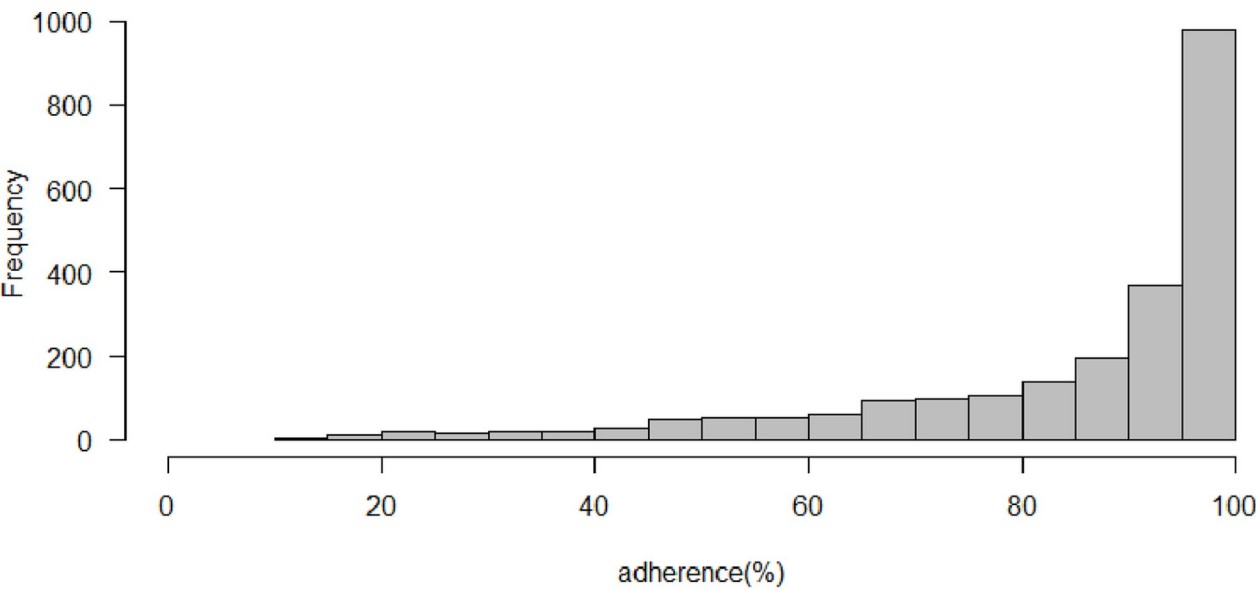

**Fig 1. Distribution of average adherence per patient in the KPNC cohort.** Average adherence was the average over all of each patient's 6-month intervals during follow-up.

(IQR:5–14). When assessing the effect of longitudinal adherence on CD4 count in the next 6-month interval, CD4 count was associated with adherence levels (regression coefficient for CD4 count = 8.54 for 10% increment in adherence, p-value <0.0001, Table 3).

When looking at temporal CD4 count changes, we found that PLWH had similar patterns over time regardless of whether they were in the lowest cumulative adherence group (adherence < 70%) or the highest cumulative adherence group (adherence > = 95%) (Fig 2A–2C). In all adherence strata, there was an initial steep increase in CD4 counts following the initiation of treatment, followed by a plateau or very shallow rise in CD4 counts.

### Effect of adherence on virologic outcomes

A total of 26045 viral loads were observed in the 2315 patients included in the final analysis. Over the entire follow-up duration, each patient had a median of 10 viral load measurements (IQR:6–16). We found that a 10% percent increase in adherence increased the odds of being virally suppressed in the next 6-month interval by 37% (OR and 95% CI: 1.37 [1.33–1.41]). Longitudinal adherence was strongly associated with a decrease in log10 viral load (regression coefficient = -0.06 for a 10% increment in adherence, p-value <0.0001, Table 3).

When looking at temporal changes in viral load, we saw similar patterns in the three adherence strata, with a steep drop in viral load after the initiation of treatment, and then a plateau (Fig 3A–3C). The average viral load remained the lowest in patients in the highest adherence group (>95%) and highest in the lowest adherence group (< 70%); on average, patients in the lowest adherence group never attained the undetectable threshold (log10 Viral load = 1.9) (Fig 3A).

### Discussion

In US clinics, PLWH can leave and re-enter care at any time point and often spend extended periods without any interaction with the healthcare system. MPR calculated using pharmacy refill data has been shown to be an accurate proxy for adherence [14–17]. Using that approach, we found an average adherence of 84.9% over the duration of follow-up in our cohort; 57.6%

**Table 2. Baseline predictors of longitudinal adherence among adult patients living with HIV initiating treatment at KPNC from 2010 to 2017\*.**

| Variables | Number of patients for each category | Crude Odds ratio (95% CI) | Adjusted Odds ratios for variables included in the final multivariable model (95%CI) |
|---|---|---|---|
| **Age (10-year increment)** | 2315 | 1.27 (1.21−1.32) | 1.25 (1.20−1.31) |
| **Race/Ethnicity:** White | 960 | 1.00 (ref) | 1.00 (ref) |
| Black | 497 | 0.57(0.49−0.66) | 0.62 (0.53−0.73) |
| Hispanic | 531 | 0.79 (0.69−0.91) | 0.90 (0.78−1.04) |
| Asian | 234 | 0.89 (0.74−1.08) | 1.02 (0.84−1.24) |
| **Sex:** Male | 2103 | 1.00 (ref) | |
| Female | 212 | 0.90 (0.75−1.09) | |
| **Transmission Risk Factor:** Men who have sex with men | 1398 | 1.00 (ref) | 1.00 (ref) |
| Injection drug users | 128 | 0.91(0.71−1.16) | 0.77 (0.58−0.96) |
| Heterosexual female | 175 | 0.93 (0.76−1.14) | 0.94 (0.82−1.27) |
| Heterosexual male | 242 | 1.10 (0.91−1.34) | 1.04 (0.88−1.33) |
| Male Bisexual | 275 | 1.02 (0.92−1.30) | 0.96 (0.88−1.23) |
| Other/ Unknown risk factor category | 97 | 0.79(0.61−1.03) | 0.73 (0.56−0.982) |
| **Baseline CD4 count:** > = 500 cells/ul | 668 | 1.00 (Ref) | |
| 350−500 cells/ul | 563 | 0.94 (0.81−1.09) | |
| 200−350 cells/ul | 468 | 0.92 (0.78−1.08) | |
| 100−200 cells/ul | 232 | 1.01 (0.83−1.23) | |
| < 100cells/ul | 330 | 1.17 (0.98−1.40) | |
| **Baseline Resistance** No | 1995 | 1.00 (ref) | |
| Yes | 320 | 0.94 (0.80−1.10) | |
| **Baseline viral load (per 1 log)** | 2269 | 1.06 (0.98−1.13) | |
| **Regimen**\*\* **EFV-based** | 603 | 1.00 | |
| ATV-based | 68 | 0.96 (0.68−1.32) | 1.03 (0.72−1.47) |
| DRV-based | 222 | 1.00 (0.83−1.22) | 1.04 (0.85−1.27) |
| DTG-based | 348 | 1.25 (1.04−1.51) | 1.35 (1.05−1.74) |
| ETR-based | 6 | 0.49 (0.20−1.23) | 0.53 (0.21−1.34) |
| EVG-based | 570 | 1.08 (0.93−1.26) | 1.15 (0.94−1.42) |
| LPV-based | 16 | 0.81 (0.41−1.58) | 1.06 (0.50−2.24) |
| NVP-based | 11 | 2.42 (1.17−4.99) | 1.94 (0.92−4.11) |
| RAL-based | 198 | 1.53 (1.23−1.90) | 1.50 (1.20−1.87) |
| RPV-based | 218 | 1.12 (0.91−1.37) | 1.22 (0.98−1.53) |
| Other | 55 | 1.26 (0.85−1.85) | 1.21 (0.83−1.75) |

\* The odds ratios were calculated using a GEE model and adjusting for year of treatment initiation.

\*\*number of observations for each drug.

of patients had an average adherence ≥95%. PLWH who were Black, young, or injecting drug users were more likely to be nonadherent.

In a previous study of this cohort, we reported that 11.7% of patients experienced virologic failure with rates of virological failure and drug resistance of 3.0 and 0.8 per 100 person-years, respectively [25]. In the current study, we showed that adherence was associated with both

**Table 3. Impact of longitudinal adherence on future viral load and CD4 count*.**

| | Coefficient for the effect of longitudinal adherence on future CD4 count (95% CI) | Coefficient for the effect of longitudinal adherence on future log10 viral load count (95% CI) |
|---|---|---|
| **Adherence (10% increment)** | 8.54 cells/mm$^3$ (5.75, 11.3) | −0.06 log copies/ml (−0.07, −0.05) |

*Adjusted for time effects by including a restricted cubic spline with 3 nodes for the time-component.

lower VL levels and higher CD4 counts. A 10% increase in adherence was associated with a 37% increase in odds of having an undetectable VL in the following 6-month interval, as well as an 8.54 increase in CD4 counts. The observed impact of adherence on CD4 counts in our cohort was relatively small and unlikely to be clinically relevant. This illustrates the importance of monitoring adherence; focusing on retention in care alone may miss gaps in treatment that have important clinical consequences.

Our estimates of adherence—average of 84%, with 57% of patients having an average adherence of at least 95%—are comparable to estimates reported in previous studies. In an early meta-analysis of 31 North American ART adherence studies, 55% of patients studied were reported to have achieved adherence levels of at least 95% [26]. However, studies included in that meta-analysis had short follow-up durations and were carried out in the previous decade [26]. In a more recently published study of a nationally representative sample of 21603 PLWH in the U.S., only 43% of patients had an average adherence ≥90% during one year of follow-up [27]. These differences in rates between that study and ours might be due to socioeconomic status differences between both study populations. Our cohort was made up of only Kaiser Permanente members whereas that study population had a large proportion of patients on Medicare and Medicaid. In addition, our study had a longer follow-up period and measured

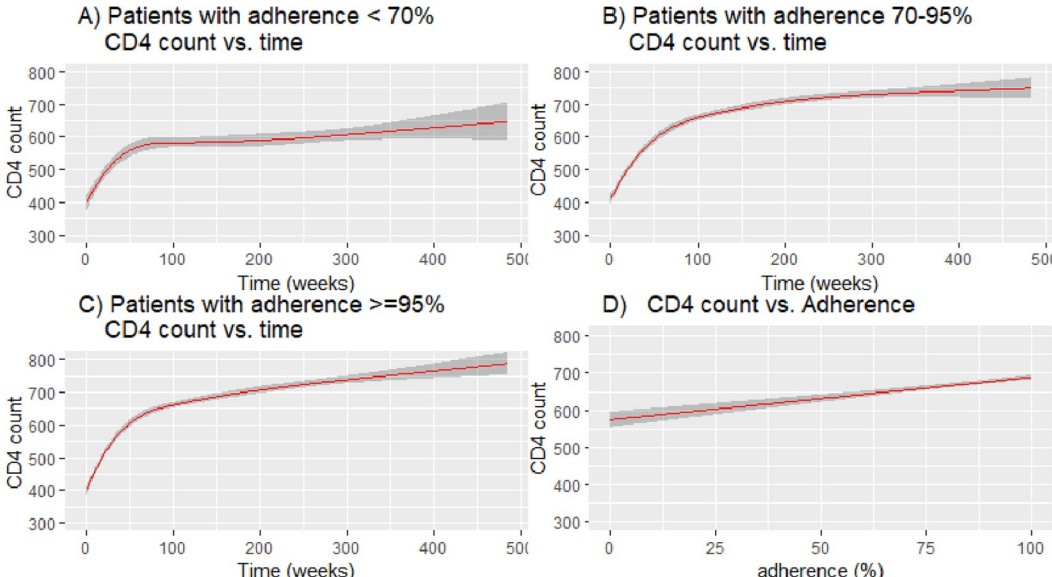

**Fig 2.** (A) Fitted model for CD4 count vs. time in weeks among patients with adherence < 70% throughout the study; (B) Fitted model for CD4 count vs. time in weeks among patients with adherence between 70% & 95% throughout the study; (C) Fitted model for CD4 count vs. time in weeks among patients with adherence > = 95% throughout the study; (D) Fitted model for CD4 count vs. adherence (%).

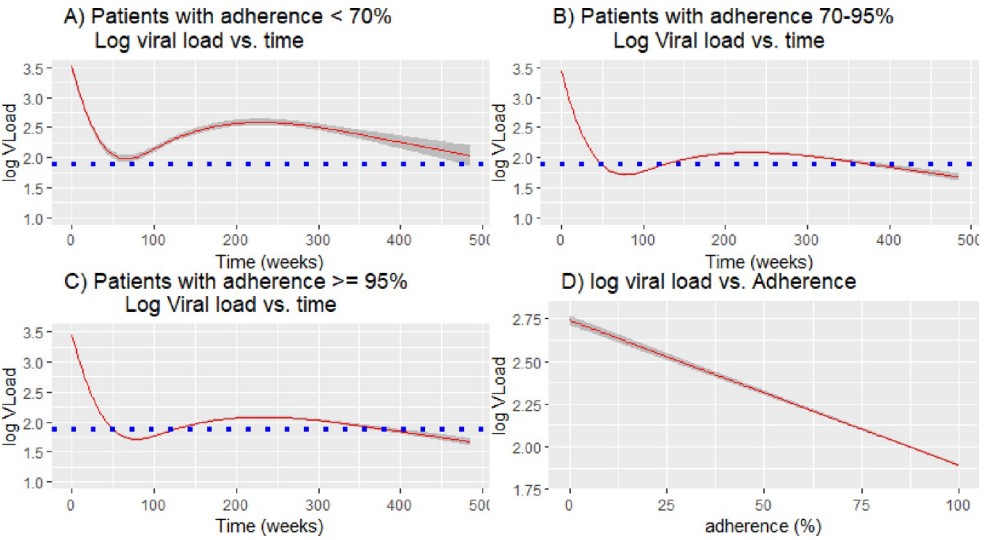

**Fig 3.** (A) Fitted model of log10 viral load vs. time in weeks among patients with adherence < 70% throughout the study;(B) Fitted model of log10 viral load vs. time in weeks among patients with adherence between 70% & 95% throughout the study; (C) Fitted model of log10 viral load vs. time in weeks patients with adherence > = 95% throughout the study; (D) Fitted model of log10 viral load vs. adherence (%). Blue dotted lines represent the lowest point of viral load detection.

adherence for multiple 6-month intervals over the entire follow-up period rather than as a single measurement. This approach is more reliable as it represents adherence as a dynamic process.

We observed different adherence rates between black PLWH and other ethnicities. Black PLWH had 1.61 times the odds of experiencing adherence less than 95% compared with white PLWH. This finding is similar to earlier studies looking at ART adherence among PLWH in US clinics [27–29]. In a US multisite 2009 cohort study, black PLWH were 1.37 times more likely to have imperfect adherence when compared with white PLWH [29]. In a 2012 study that pooled data from 13 different US studies, the odds of having perfect adherence for black PLWH was 0.60 times that of white PLWH after accounting for education, income, depression, and substance use [28]. Although we did not adjust for those variables in our study, our odds ratio estimate is extremely similar, as 1/1.61 = 0.62.

Although most previous adherence studies were carried out before the emergence of integrase inhibitors, it is alarming to note these ethnic differences in adherence persist to this day. This is even more concerning given that HIV/AIDS disproportionally affects minorities in the United States [29, 30]. Although diagnosis, time to treatment initiation, and retention in care have improved, disparities in adherence rates remain [28–31].

We also found sub-optimal adherence rates among injecting drug users in our study highlighting the different treatment access challenges that this sub-population faces and underscoring the possible need for special treatment strategies for this population [32, 33]. PLWH receiving a dolutegravir or raltegravir-containing regimen had higher adherence rates compared with the reference efavirenz-based regimen group. This could be explained by differences in tolerability of these regimens, in particularly the central nervous system side effects that have been reported in PLWH using Efavirenz-based regimens [34, 35].

Many studies have established a link between adherence and poor virologic outcomes [4, 8, 9]. We observed similar patterns in our study. On average, patients with a mean 6-month adherence <70% did not attain undetectable viral load levels. In our study, although

significant, the relationship between adherence and CD4 count in the next 6-month interval was weak in magnitude (8.54 cells/mm3 increase per 10% increase in adherence), which could be explained by the fact that even patients with modest adherence experienced a dramatic rebound in CD4 counts in the first year after treatment initiation. The weak relationship between CD4 counts and adherence may also be due to the fact that nonadherent patients are less likely to show up for testing. Hence, this could bias our results towards the null as we are not capturing all the dynamic dips in CD4 counts. Finally, because our CD4 counts model was a lagged model assessing the impact of adherence on CD4 counts in the next 6-month interval, the timing of the model may not perfectly capture the impact of adherence on CD4 counts dynamics.

Our study has limitations. First, because we did not have access to full patients' records, it was not possible to ascertain the effect of other co-morbidities on adherence and CD4 counts. Therefore, the presence of unmeasured confounding cannot be completely ruled out. In addition, although MPR is a good proxy for adherence, it is impossible to ascertain that a PLWH took their medication after collecting it from the pharmacy. Finally, the low percentage of female patients in the cohort and the fact that all PLWH in this cohort were insured restricts the generalizability of this study to only populations with similar characteristics.

In conclusion, our study shows that despite large improvements in retention in care, demographic disparities in adherence to ART and loss-to-follow-up still persist. African American and young PLWH are more likely to be non-adherent to ART. Furthermore, young PLWH and those with advanced disease stage at baseline were more prone to be out of continuous care. In addition, our study confirmed the strong association between adherence to ART and viral suppression but could only establish that there was a weak association between adherence and CD4 count. To our knowledge, this is one of the few large recent studies assessing the impact of adherence on CD4 count in clinic settings. These findings reaffirm the importance of adherence and retention in care and underline the need for tailored patient-centered HIV Care Models.

## Author Contributions

**Conceptualization:** Christophe T. Tchakoute, Soo-Yon Rhee, C. Bradley Hare, Robert W. Shafer.

**Data curation:** Christophe T. Tchakoute, Soo-Yon Rhee, Robert W. Shafer, Kristin Sainani.

**Formal analysis:** Christophe T. Tchakoute, Kristin Sainani.

**Funding acquisition:** Soo-Yon Rhee.

**Investigation:** C. Bradley Hare, Robert W. Shafer, Kristin Sainani.

**Methodology:** Christophe T. Tchakoute, Kristin Sainani.

**Project administration:** Robert W. Shafer.

**Resources:** Robert W. Shafer.

**Supervision:** Robert W. Shafer.

**Validation:** Christophe T. Tchakoute, Robert W. Shafer, Kristin Sainani.

**Visualization:** Kristin Sainani.

**Writing – original draft:** Christophe T. Tchakoute.

**Writing – review & editing:** Robert W. Shafer, Kristin Sainani.

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
