## [Decision Letter · Decision Letter 0]

2 Sep 2021

PONE-D-21-23391

Adherence to contemporary Antiretroviral Treatment Regimens and impact on immunological and virologic outcomes in a US healthcare system

PLOS ONE

Dear Dr. Toukam Tchakoute,

Thank you for submitting your manuscript to PLOS ONE. After careful consideration, we feel that it has merit but does not fully meet PLOS ONE’s publication criteria as it currently stands. Therefore, we invite you to submit a revised version of the manuscript that addresses the points raised during the review process.

We look forward to receiving your revised manuscript.

Kind regards,

Vincent C Marconi

Academic Editor

PLOS ONE

Journal Requirements:

2. Please clarify whether the data obtained from the KPNC research database were anonymized or deidentified before use. If not, please provide information regarding informed consent from the subjects.

“: S.Y.R. and R.W.S. were supported in part by the National Institute of Allergy and Infectious Diseases (NIAID) of the National Institute of Health (NIH) (award number AI136618).“

“Funding: S.Y.R. and R.W.S. were supported in part by the National Institute of Allergy and Infectious Diseases (NIAID) of the National Institute of Health (NIH) (award number AI136618).”

We note that you have provided funding information within the Acknowledgements. Please note that funding information should not appear in the Acknowledgments section or other areas of your manuscript. We will only publish funding information present in the Funding Statement section of the online submission form.

“: S.Y.R. and R.W.S. were supported in part by the National Institute of Allergy and Infectious Diseases (NIAID) of the National Institute of Health (NIH) (award number AI136618).”

6. Please amend your manuscript to include your abstract after the title page.

Reviewers' comments:

Reviewer's Responses to Questions

**Comments to the Author**

1. Is the manuscript technically sound, and do the data support the conclusions?

Reviewer #1: No

Reviewer #2: Yes

2. Has the statistical analysis been performed appropriately and rigorously? 

Reviewer #1: No

Reviewer #2: Yes

3. Have the authors made all data underlying the findings in their manuscript fully available?

Reviewer #1: Yes

Reviewer #2: Yes

4. Is the manuscript presented in an intelligible fashion and written in standard English?

Reviewer #1: Yes

Reviewer #2: Yes

5. Review Comments to the Author

Reviewer #1: This paper is timely and needed.

I recommend that this paper be broken into two papers, one that focuses on loss to follow-up and one focusing on the GEE. Currently, this paper is too ambitious, and it has lost focus. There is too much here for the number of tables and figures you are limited to.

The paper is not organized and is difficult to read. There is too much going on.

The paper needs to be revised to reconcile the numbers in the text with the numbers in the tables. There are entire sections that utilize demographic frequencies but do not match the table one.

I agree with the use of the GEE, and appreciate the novelty in this area; however, I have concerns about the way the analyses are performed. You, for example, estimate adherence ORs from viral load which is interesting; however, you also reverse that model to predict viral load from adherence and that is not appropriate. Models are created for prediction/association with explanatory variable and repones variables if I say, for example, smoking explains lung cancer it would not make sense to say lung cancer explains smoking.

You report linear regression betas by referring the reader to a graph, but you should have a table with the betas listed. It is vague exactly what explantory variables are in your models.

Your Table One needs to be formatted and report standard measures. For categorical you must report frequency and relative frequency. For continuous non-skewed data, I want to see mean and standard deviation and if the data is not normally distributed, I want median and IQR.

You need to clarify the regimen that each patient was taking during the study period, you mention EFV and DTG but the standard of treatment in the USA is a triple regimen in a single pill. Did you mean EFV and DTG based regimens? I would immediately question any result with a subject on either of regimens as a monotherapy.

Reviewer #2: Shafer et al followed a cohort of 2135 patients living with HIV (PLWH) in this study from 2010 to 2017 in California to assess longitudinal adherence and its impact on immunological and virological outcomes in patients initiating antiretroviral therapy (ART). They utilized pharmacy records and calculated the medication possession ratio (MPR) within multiple 6-month intervals. The authors followed PLWH for a median time of 210.8 weeks and 179(7.7%) were lost to follow up. The mean adherence per person was 84.9% and 57.6% of PLWH had an average adherence level > or equal to 95% over the entire follow-up duration. After controlling for year of treatment initiation, four baseline variables remained significantly associated with optimal adherence: age (a ten-year increase in age was associated with a 25% increase in odds of optimal adherence), ethnicity (Blacks had significantly lower odds of optimal adherence when compared to Whites), transmission risk factors (Injecting drug users were less likely to achieve optimal adherence when compared with the MSM reference group), and first-line regimen (PLWH on integrase strand transfer inhibitors were more likely to achieve optimal adherence when compared with PLWH on NNRTs). A 10% increase in adherence increased the odds of being virally suppressed by 37% in the next 6-month interval and an 8.54 increase in CD4 counts. The observed impact of adherence on CD4 counts in the cohort was relatively small and the authors concluded that it is unlikely to be clinically significant. The median CD4 count of the cohort at baseline was 373.

The overall retention in this cohort was high and the study illustrates the importance of using surrogate markers (MPR) of ARVs adherence monitoring and its impact on virological outcomes.

Few questions/comments to the authors:

1. Results section: When assessing the effect of longitudinal adherence on CD4 count in the next 6-month interval, CD4 count was associated with adherence levels. When looking at temporal CD4 count changes, the authors found that PLWH had similar patterns over time regardless of their ARVs adherence group. There was an initial steep increase in CD4 counts following the initiation of treatment, followed by a plateau or very shallow rise in CD4 counts. Were the authors able to assess the CD4 count changes relative to baseline CD4 count in PLWH (e.g. those with CD4 count <200 compared to those >200)?

2. Results section: The authors concluded that one of the baseline variables that remained significantly associated with optimal adherence after controlling for year of treatment initiation was first-line regimen: PLWH on INSTIs were more likely to achieve optimal adherence when compared with PLWH on NNRIs. Do the authors have the data on how many PLWH were initiated on INSTIs during the study and how did that change during the study period? And were there any differences between different ethnic groups (White, Black, etc.)?

3. Discussion section: The authors conclude that ethnic differences persist and the fact that HIV/AIDS disproportionately affects minorities in the United States. Similar studies found that adherence for Black PLWH was lower when compared to Whites PLWH. Do the authors have data on the prevalence of mental health illness in this cohort including substance use (e.g. alcohol, cocaine, tobacco, etc.) at baseline?

4. Overall: The authors do not comment on this but did they implement RAPID or SAME day ART especially towards the latest part of the study?

6. PLOS authors have the option to publish the peer review history of their article (what does this mean?). If published, this will include your full peer review and any attached files.

Reviewer #1: No

Reviewer #2: No

---

## [Author Response · Author response to Decision Letter 0]

18 Dec 2021

Response to Reviewer 1

I recommend that this paper be broken into two papers, one that focuses on loss to follow-up and one focusing on the GEE. Currently, this paper is too ambitious, and it has lost focus. There is too much here for the number of tables and figures you are limited to.

Yes, we agree that the inclusion of the material on “loss to follow up” is confusing. This was included because we were considering “loss to follow up” as an extreme version of non-adherence. However, we appreciate the reviewer drawing our attention to the fact that this added analysis distracts from the main focus of the paper, which is drug adherence. Per the reviewer’s request, we have now removed this material, including Table 3.

The paper is not organized and is difficult to read. There is too much going on.

Thank you for pointing out that we attempted to include too much material in this paper. We have now removed the “loss to follow up” analysis, which we believe improves the clarity of the paper. We also agree that some of the prose, particularly methods and results, lacked clarity. We have now edited these two sections to improve clarity. 

The paper needs to be revised to reconcile the numbers in the text with the numbers in the tables. There are entire sections that utilize demographic frequencies but do not match the table one.

Thank you for catching these errors. We indeed had two errors in the text, which have now been corrected. We also removed the statement that “92% of PLWH had undetectable VL at the end of follow-up” from the first paragraph of the Results to keep the focus of this first paragraph only on baseline demographics. We have also edited the table and text for clarity, such as adding N’s to the table in addition to percentages and editing the title. 

-I agree with the use of the GEE, and appreciate the novelty in this area; however, I have concerns about the way the analyses are performed. You, for example, estimate adherence ORs from viral load which is interesting; however, you also reverse that model to predict viral load from adherence and that is not appropriate. Models are created for prediction/association with explanatory variable and repones variables if I say, for example, smoking explains lung cancer it would not make sense to say lung cancer explains smoking.

We appreciate the reviewer drawing our attention to the fact that our written description of our GEE models lacked clarity. To be clear, the two models are not the same models in reverse. The first model (results presented in Table 1, described in part a of the statistical methods section), models the BASELINE viral load on subsequent, longitudinal adherence to drugs. This model is answering the question of whether starting with more uncontrolled disease is a predictor of non-adherence. The second GEE model (described in parts b and c of the statistical methods section) models the effect of adherence on SUBSEQUENT, LONGITUDINAL changes in viral load. The predictor is adherence in each six-month interval and the outcome is viral load in each SUBSEQUENT six-month interval. The model is answering the question of whether one’s adherence to antiviral medications affects one’s viral load over time. We agree that the statistical methods section was confusing and did not make these distinctions clear. We have now edited the statistical methods section to improve clarity. 

-You report linear regression betas by referring the reader to a graph, but you should have a table with the betas listed. It is vague exactly what explanatory variables are in your models.

This is an excellent point. We have now added a table 3 that gives the beta coefficients for the models. We have also added a footnote on the table indicating what was adjusted for in those models. We believe that this is very helpful to the reader. 

-Your Table One needs to be formatted and report standard measures. For categorical you must report frequency and relative frequency. For continuous non-skewed data, I want to see mean and standard deviation and if the data is not normally distributed, I want median and IQR.

We apologize that Table One was confusing and poorly formatted. All of the continuous values are given as median and IQR’s already. But we were indeed missing frequencies/N’s for some categorical variables. That has now been fixed. We have also improved the title of the table for clarity.

-You need to clarify the regimen that each patient was taking during the study period, you mention EFV and DTG but the standard of treatment in the USA is a triple regimen in a single pill. Did you mean EFV and DTG based regimens? I would immediately question any result with a subject on either of regimens as a monotherapy.

We appreciate the reviewer drawing our attention to the lack of clarity in our description of regimens. These are EFV and DTG-based triple regimens 2. We added terms to table 2 and to lines 214 & 215 on page 10 to reflect these changes. 

 Response to Reviewer 2

1. Results section: When assessing the effect of longitudinal adherence on CD4 count in the next 6-month interval, CD4 count was associated with adherence levels. When looking at temporal CD4 count changes, the authors found that PLWH had similar patterns over time regardless of their ARVs adherence group. There was an initial steep increase in CD4 counts following the initiation of treatment, followed by a plateau or very shallow rise in CD4 counts. Were the authors able to assess the CD4 count changes relative to baseline CD4 count in PLWH (e.g. those with CD4 count <200 compared to those >200)?

We appreciate the reviewer’s question on CD4 count changes relative to baseline CD4 count in PLWH. After adjusting for baseline regimens, age, baseline viral load and year of treatment of initiation, patients that had baseline CD4 count under 200 cells/mm3 had a significantly lower CD4 count at the end of the cohort (-354 cells/mm3 ) when compared to those who had baseline CD4 count over 200 cells/mm3.

2. Results section: The authors concluded that one of the baseline variables that remained significantly associated with optimal adherence after controlling for year of treatment initiation was first-line regimen: PLWH on INSTIs were more likely to achieve optimal adherence when compared with PLWH on NNRIs. Do the authors have the data on how many PLWH were initiated on INSTIs during the study and how did that change during the study period? And were there any differences between different ethnic groups (White, Black, etc.)?

We appreciate this relevant question on temporal changes in baseline treatment regimens . We previously reported on yearly changes in initiated treatment in a recently published manuscript1. See Figure below describing yearly distribution of the different baseline regimens in this cohort. In addition, regarding potential ethnic different in treatment initiated during each year, we found no statistically significant differences between the different ethnic groups in this cohort.

3. Discussion section: The authors conclude that ethnic differences persist and the fact that HIV/AIDS disproportionately affects minorities in the United States. Similar studies found that adherence for Black PLWH was lower when compared to Whites PLWH. Do the authors have data on the prevalence of mental health illness in this cohort including substance use (e.g. alcohol, cocaine, tobacco, etc.) at baseline?

We appreciate this question on prevalence of mental health illness and substance use. Recent studies reported prevalence of 4.3% for any psychiatric diagnosis2 and an opioid use prevalence of 8% in this cohort.3

4. Overall: The authors do not comment on this but did they implement RAPID or SAME day ART especially towards the latest part of the study?

 We appreciate the reviewer’s question on the timing of ART initiation. On average, RAPID/SAME day ART was not implemented in this patient population. ART was generally initiated until genotypic resistance testing results were available. We added the following statement “With few exceptions, ART was begun after the results of genotypic resistance testing were available” on line 89 to reflect this point.

References

1. Rhee SY, Clutter D, Hare CB, Tchakoute CT, Sainani K, Fessel WJ, Hurley L, Slome S, Pinsky BA, Silverberg MJ, Shafer RW. Virological Failure and Acquired Genotypic Resistance Associated With Contemporary Antiretroviral Treatment Regimens. In Open forum infectious diseases 2020 Sep (Vol. 7, No. 9, p. ofaa316). US: Oxford University Press.

2. Marcus JL, Hurley LB, Krakower DS, Alexeeff S, Silverberg MJ, Volk JE. Use of electronic health record data and machine learning to identify candidates for HIV pre-exposure prophylaxis: a modelling study. The lancet HIV. 2019 Oct 1;6(10):e688-95.

3. Silverberg MJ, Ray GT, Saunders K, Rutter CM, Campbell CI, Merrill JO, Sullivan MD, Banta-Green C, Von Korff M, Weisner C. Prescription long-term opioid use in HIV-infected patients. The Clinical journal of pain. 2012 Jan;28(1):39.

---

## [Decision Letter · Decision Letter 1]

26 Jan 2022

Adherence to contemporary Antiretroviral Treatment Regimens and impact on immunological and virologic outcomes in a US healthcare system

PONE-D-21-23391R1

Dear Dr. Toukam Tchakoute,

We’re pleased to inform you that your manuscript has been judged scientifically suitable for publication and will be formally accepted for publication once it meets all outstanding technical requirements.

Kind regards,

Vincent C Marconi

Academic Editor

PLOS ONE

Additional Editor Comments (optional):

Reviewers' comments:

Reviewer's Responses to Questions

**Comments to the Author**

1. If the authors have adequately addressed your comments raised in a previous round of review and you feel that this manuscript is now acceptable for publication, you may indicate that here to bypass the “Comments to the Author” section, enter your conflict of interest statement in the “Confidential to Editor” section, and submit your "Accept" recommendation.

Reviewer #1: All comments have been addressed

2. Is the manuscript technically sound, and do the data support the conclusions?

Reviewer #1: Yes

3. Has the statistical analysis been performed appropriately and rigorously? 

Reviewer #1: Yes

4. Have the authors made all data underlying the findings in their manuscript fully available?

Reviewer #1: Yes

5. Is the manuscript presented in an intelligible fashion and written in standard English?

Reviewer #1: Yes

6. Review Comments to the Author

Reviewer #1: I appreciate the work that has gone into your manuscript and its revisions. The paper is much more clear now and the statistics are appropriate and well defined.

7. PLOS authors have the option to publish the peer review history of their article (what does this mean?). If published, this will include your full peer review and any attached files.

Reviewer #1: **Yes: **Johnathan A. Edwards

---

## [Editor Report · Acceptance letter]

2 Feb 2022

PONE-D-21-23391R1 

Adherence to contemporary antiretroviral treatment regimens and impact on immunological and virologic outcomes in a US healthcare system 

Dear Dr. T. Tchakoute:

I'm pleased to inform you that your manuscript has been deemed suitable for publication in PLOS ONE. Congratulations! Your manuscript is now with our production department. 

Kind regards, 

on behalf of

Dr. Vincent C Marconi 

Academic Editor

PLOS ONE